# Biochemical and Anthropometric Parameters for the Early Recognition of the Intrauterine Growth Restriction and Preterm Neonates at Risk of Impaired Neurodevelopment

**DOI:** 10.3390/ijms241411549

**Published:** 2023-07-17

**Authors:** Maria Cristina Aisa, Benito Cappuccini, Alessandro Favilli, Alessandro Datti, Vincenza Nardicchi, Giuliana Coata, Sandro Gerli

**Affiliations:** 1Department of Surgical and Biomedical Sciences, Section of Obstetrics and Gynecology, University of Perugia, 06129 Perugia, Italy; alessandro.favilli@unipg.it (A.F.); giuliana.coata@unipg.it (G.C.); sandro.gerli@unipg.it (S.G.); 2GeBiSa, Research Foundation, 06129 Perugia, Italy; benitocappuccini@hotmail.com (B.C.); vincenzanardicchi@mac.com (V.N.); 3Centre of Perinatal and Reproductive Medicine, University of Perugia, 06129 Perugia, Italy; 4Department of Agricultural, Food and Environmental Sciences, University of Perugia, 06121 Perugia, Italy; alessandro.datti@unipg.it

**Keywords:** impaired neurodevelopment, IUGR, preterm, S100B, Tau, NGF, cerebral volumes

## Abstract

Background: S100B and Tau are implicated with both brain growth and injury. Their urinary levels in 30-to-40-day-old full-term, preterm, IUGR, and preterm-IUGR subjects were measured to investigate their possible relationship with future delayed neurodevelopment. Methods: Values were related to the neuro-behavioral outcome at two years of age, as well as to brain volumes and urinary NGF assessed at the same postnatal time point. Results: Using the Griffiths III test, cognitive and motor performances were determined to establish subgroups characterized by either normal or impaired neuro-behavior. The latter included preterm, IUGR, and preterm-IUGR individuals who exhibited significantly higher and lower S100B and Tau levels, respectively, along with markedly reduced cerebral volumes and urinary NGF, as previously demonstrated. Contrary to NGF, however, Tau and S100B displayed a weak correlation with brain volumes. Conclusions: Delayed cognitive and motor performances observed in two-year-old preterm and IUGR-born individuals were also found to be associated with anomalous urinary levels of S100B and Tau, assessed at 30–40 days of the postnatal period, and their changes did not correlate with brain growth. Thus, our data suggests that, in addition to cerebral volumes and NGF, urinary S100B and Tau can also be considered as valuable parameters for the early detection of future neurodevelopmental abnormalities.

## 1. Introduction

Intrauterine growth restriction (IUGR) and preterm births are associated with a increased risk of impaired cognitive and motor performances or neuro and psychiatric diseases in later life [1,2,3]. Both conditions are also correlated with a reduction in brain volumes, showing the strong relationship between cerebral growth and neuro-disorders [1,2,3]. The early identification of neonates and children at a higher risk of defective neurodevelopment or neurological disability is very important, as they may necessitate prompt interventions to improve their developmental issues [4,5]. Therefore, the Food and Drug Administration, the European Medicine Agency, and the National Institutes of Health promoted research projects towards the identification and validation of new markers, along the development of neuroprotective approaches for individuals at elevated risk [6].

To date, however, no well-established, early indicators of the long-term impairments of neurodevelopment are currently available, particularly with regard to the initial phase of the postnatal period.

Recently, the levels of the 3D echo cerebral volumes and urinary NGF were found to be significantly lower in 30–40 day-old preterm, IUGR, and preterm-IUGR neonates when compared to full-term subjects, and to strictly correlate with defective neurodevelopmental outcomes at two years of age [1,2,3]. Therefore, their role as valuable tools for the early prediction of long-term neurological damage has been suggested [1,2,3].

S100B and Tau are two proteins which are expressed in high abundance in the central nervous system, where they are involved in many neuro-, physio-, and pathological processes.

S100B is an acidic calcium-binding protein belonging to the EF motif-hand family and is characterized with a low molecular weight (of about 21 kD). It is mainly concentrated in the glial cells and in restricted neuron subpopulations [7,8]. In the nervous system, S100B plays a biphasic role [7,8]. At physiological concentrations (nanomolar), it exhibits neurotrophic effects. It regulates several cellular functions (i.e., cell growth, cell structure, and energy metabolism), stimulates neurite growth, and promotes neuronal survival [9]. At higher levels (micromolar level), S100B displays neurotoxic effects, including the impairments in synaptic plasticity and spatial learning, and the apoptosis of astrocytes and their neighboring neurons [7,8] induced by the stimulation of ROS and NOS production [10]. Its presence in the extracellular space may be due to its leakage from damaged cells [11], or to active secretion, thus functioning as an extracellular signaling molecule with concentration-dependent effects [11,12]. S100B is quickly released from the brain into the blood when the blood–brain barrier (BBB) is disrupted [13]. Furthermore, this protein has a half-life of 2 h, and is mainly eliminated by the kidney (98%) [14]. The S100B protein has been measured in several biological fluids (CSF, blood, urine, and amniotic fluid), where its overexpression has been primarily linked to active brain injuries and the pathogenesis of neural disorders (i.e., Alzheimer’s disease, Parkinson’s disease, amyotrophic lateral sclerosis, and multiple sclerosis), suggesting a diagnostic potential [11,12,15]. In adults, S100B increases have also been found in the blood and urine during mental and physical activity, under stress conditions, and in mood disorders [16,17,18]. During the perinatal phase, higher amounts of this molecule in the urine and blood of infants have been observed to correlate with insult conditions, such as intraventricular hemorrhage or hypoxic ischemic encephalopathy [19,20,21,22,23]. However, an inverse and strong correlation between S100B, from the cord blood and urine of neonates at birth, and gestational age (GA) has suggested that increased amounts of this protein in preterm subjects could be related to its neurotrophic function [24,25], and therefore be indicative of the brain maturation processes that are particularly active [25]. Increased levels of S100B were also demonstrated in urine samples from IUGR newborns, with the highest values shown in those with abnormal neurological outcomes [26].

Tau, the tubulin-associated unit protein, is a highly soluble microtubule assembly molecule present in both the neurons of the central nervous system and oligodendrocytes [27,28]. It can perform both trophic and toxic activities. In the adult neuron, it is primarily located in the axon inducing the stabilization of microtubules and the cytoskeletal network [29] and the maintenance of axonal transport [30]. With its dendritic fraction, it participates in the regulation of synaptic plasticity [31]. It is strongly expressed in the developing brain participating in fundamental events, including the establishment of neuronal polarity and migration during embryonic development, the functional maturation and survival of newborn neurons, the selectivity of neuronal death following stress, and neuronal responses to external stimuli [32,33,34]. Post-translational modifications of Tau may impair its localization and functions, particularly their involvement in neuronal migration and synapse formation, resulting in a spectrum of neurological diseases, such as schizophrenia, intellectual disability, autism [35,36,37,38,39], and tauopathies [40]. According to its anatomical and ultrastructural localization, Tau was recognized as a marker of axonal disruption, and its release into the blood and cerebrospinal fluid has been indicated as a sign of neurotrauma and a predictor of injury severity of the central nervous system (i.e., in traumatic brain injury, cerebral stroke, Alzheimer’s disease, and in hypoxic ischemic encephalopathy injury in neonates [41,42,43,44,45]).

In the present study, we evaluated possible associations between S100B, Tau, and delayed neurodevelopment later in life in preterm and IUGR subjects. In particular, the urinary levels of S100B and Tau, measured at 30–40 days of postnatal period, were investigated for their correlation with the motor and behavioral outcomes of the same subjects after two years of follow-up. We also considered a possible relationship of these proteins with other variables measured at the same postnatal time point, such as the cerebral volume (i.e., whole brain volume, WB: thalamus volume, TV; cerebellum volume, CV; and frontal cortex volume, FCV) to better establish their association with brain growth, and urinary NGF, as NGF and Tau activities have been shown to correlate during early neuronal development or in neuro-diseases [46,47].

The prognostic accuracy of the parameters investigated here and the possible predictive multivariate logistic regression models for the recognition of neonates at risk of impaired neurodevelopment at two years were also considered.

## 2. Results

### 2.1. Variables Assessed at 30–40 Days of the Postnatal Period and Related Correlation Studies in the Whole Population

Levels of S100B, Tau, NGF, brain volumes, and BW, measured at 30–40 days of the postnatal period in the whole population, are reported in Table 1.

Biochemical and anthropometric parameters were assessed for correlation. With regard to cerebral volumes, Tau and S100B exhibited positive and negative weak relationships, respectively, whereas NGF displayed a strong correlation. In particular, the Spearman rho values of WBV, TV, FCV, and CV were 0.232 (*p* = 0.053), 0.318 (*p* = 0.08), 0.348 (*p* = 0.055), and 362 (*p* = 0.045) vs. Tau; −0.370 (*p* = 0.04), −0.403 (*p* = 0.024), −0.448 (*p* = 0.011), and −0.387 (*p* = 0.03) vs. S100B, and 0.806 (*p* < 0.001); 0.816 (*p* < 0.001), 0.804 (*p* < 0.001), and 0.801 (*p* < 0.001) vs. NGF, respectively.

The three urinary compounds which were weakly or not significantly correlated with one another displayed the following rho values: Tau vs. NGF: 0.227 (*p* = 0.06); Tau vs. S100B: −0.441 (*p* = 0.005); and S-100 vs. NGF: −0.351 (*p* = 0.059).

### 2.2. Variables Assessed at 30–40 Days of the Postnatal Period in Full-Term, Preterm, IUGR, and Preterm-IUGR Subjects

Descriptive statistics, followed by multiple comparison analyses of the variables measured at 30–40 days post-birth, were then assessed in full-term, preterm, IUGR and preterm-IUGR neonates (Table 2).

Except for Tau in the IUGR subjects, the levels of urinary S-100B, Tau, and NGF significantly varied in preterm, IUGR, and preterm-IUGR newborns when compared to their full-term references. More specifically, Tau and NGF decreased while S100B increased. A greater reduction in Tau and NGF was found in the preterm-IUGR category (Table 2).

In agreement with previous data [1,2,3], the full-term and the preterm-IUGR neonates exhibited significantly larger and smaller cerebral volumes, respectively (Table 2).

### 2.3. S100B, Tau, NGF, and Regional Brain Volumes in the Normal and Abnormal Neurodevelopment Subgroups

The variabilities in the biochemical and anthropometric parameters were then related to the neurodevelopment outcome after two years. To this end, normal and abnormal neurodevelopment subgroups were also compared with reference to neonatal categories.

The subjects affected by abnormal neurodevelopment displayed significantly reduced levels of Tau, NGF, and cerebral volumes, while S100B was significantly increased (Figure 1 and Figure 2).

Subjects born full-term exhibited generally higher values of Tau, NGF, and cerebral volumes compared to the other categories in both subgroups (Figure 3 and Figure 4). The differences were found to be statistically significant vs. all the categories with abnormal outcome (Figure 3 and Figure 4). On the contrary, preterm-IUGR born subjects showed lower or significantly lower values in both subgroups (Figure 3 and Figure 4).

In the case of S100B, full-term subjects displayed lower urinary levels compared to preterm, IUGR, and preterm-IUGR individuals, with differences being statistically significant vs. the abnormal neurodevelopment subgroup categories (Figure 3). Within the single subgroups, no significant differences were found between the preterm, IUGR, and preterm-IUGR born infants (Figure 3).

Moreover, comparing the same category in the two subgroups, a significant increase in Tau, NGF, and cerebral volume was observed in the normal neurodevelopment subgroup, while S100B significantly decreased (Figure 3 and Figure 4).

As expected, the BW was significantly lower in the abnormal neurodevelopment subgroup (normal neurodevelopment subgroup: median 1989 g, IQR 1835–2367 g, and mean ± SD 2053 ± 280 g; abnormal neurodevelopment subgroup: median 1675 g, IQR 1520–1774 g, and mean ± SD 1643 ± 221 g) [1,2,3].

### 2.4. Diagnostic Accuracy in Determining the Negative Outcome at Two Years

The diagnostic accuracy of the variables measured in this study and the possible predictive multivariate logistic regression models for the recognition of neonates at risk of impaired neurodevelopment at two years were preliminary evaluated by the assessment of the areas under the ROC curves (AUCs) and bivariate and multivariate logistic regression analysis. The latter were performed across the whole population, whereas the ROC curve assessment was carried out by either including or excluding the full-term category.

In agreement with previous findings [1,2,3], NGF and regional volumes revealed an excellent accuracy that was significantly higher compared to Tau and S100B. By excluding the full-term category, the AUCs were as follows; NGF, 0.920 (*p* < 0.001); WBV, 0.941 (*p* < 0.001); TV, 1.000 (*p* < 0.001); FCV, 0.961 (*p* < 0.001); CV, 0.968 (*p* <0.001); Tau, 0.808 (*p* < 0.001); and S100B, 0.786 (*p* < 0.001).

Inclusion of the full-term category boosted the AUC values (Table 3) with parameters ranking for accuracy as follows: TV ≈ CV ≈ FCV ≈ WBV > NGF > Tau (*p* < 0.01) and S100B (*p* < 0.01).

Data of bivariate logistic regression analysis indicated that all variables under investigation were significantly associated with neurodevelopment impairment at two years of age (Table 3). Multivariate logistic regression models only included the biochemical parameters since all cerebral volumes singularly exhibited an excellent accuracy.

The accuracy of the combination of Tau and NGF was excellent and comparable to those of the cerebral volumes (Table 3).

## 3. Discussion

The present study validates our previous findings and suggestions by our group about the possible role of 3D echo cerebral volumes (i.e., WBV, TV, FCV, and CV) and urinary NGF for the prediction of impaired neurodevelopment in later life in preterm and IUGR infants [1,2,3]. Here, we extended the work to also show the diagnostic relevance of two proteins, S100B and Tau, which are highly expressed in the central nervous system.

Using a homogeneous and highly selected population of full-term, preterm, IUGR and preterm IUGR neonates, which did not significantly differ for gender and GA, we found that, similar to previous data [1,2,3], the levels of urinary NGF and cerebral volumes measured at 30–40 days after birth were significantly reduced in the subjects affected by delayed neurodevelopment at two years of age. The same was also observed for urinary Tau, whereas S100B showed a significant increase.

Similar patterns of urinary S100B were previously reported in preterm and IUGR neonates, with higher values observed in the IUGR subjects with abnormal neurological outcomes [26]. In the preterm neonates, however, S100B was demonstrated to be inversely and strongly correlated with the GA [25]. Since BW varies according to the GA, and as this study focused on newborns with a sex ratio close to 50%, and who did not significantly differ in the GA, S100B variability appears therefore to correlate better with the BW than to changes in the GA. In our study population, indeed, the two subgroups differed in terms of the BW.

Larger concentrations of S100B in the biological fluids have also been described in brain damage conditions associated with several pathological events at birth [6,19,20,21,22,23], also together with concurrent increased Tau levels [48]. In the serum of infants with early phase bilirubin encephalopathy, these two proteins were overexpressed and strongly correlated [48]. The same trend was observed in the cerebrospinal fluid and/or serum of infants with acute encephalopathy or autism [49,50]. Tau, alone, was higher in IUGR neonates, and was correlated with severity grade and neurodevelopmental retardation assessed at 9 months of age [45]. In our study, we found decreased levels of urinary Tau in subjects with delayed neurodevelopment. However, our population did not include neonates affected by the above pathological conditions.

The significance of the anomalous levels of S100B and Tau in the neonates with negative outcomes and different variability trends is still elusive. However, given that increased Tau levels have been indicated as predictors of neurotrauma and injury severity of the central nervous system [41,42,43,44,45], it is therefore likely that subclinical brain damage in the above conditions concerns only the glial component of the brain or, alternatively, is ruled out. In this case, an asynchronous regulatory mechanism of expression of these two molecules could be speculated, reflecting differential responses of the neurons and glia. As in the human fetal brain, Tau is expressed during neuronal maturation throughout brain development [34], and as S100B under physiological concentrations can stimulate survival, differentiation, and proliferation of neurons [51,52], it is therefore conceivable that the deficient neurodevelopment observed in the IUGR and preterm neonates underlies a mechanism which involves both the impairment of neurons, in terms of their number and maturation, and the active response of glia, with the overexpression and secretion of S100B. The observed increase in S100B may be also linked to a possible inflammatory state [15], and in IUGR neonates, to metabolic stress or an “early” glucose intolerance and insulin resistance which typically occurs in this condition [53,54,55]. In all the suggested hypotheses, however, it should be considered how S100B, as well as Tau and NGF, cross the BBB [13]. Although in some cases it has been criticized [56,57], the increased levels of S100B in the blood have usually been considered as an indicator of BBB damage. Thus, the BBB in IUGR and preterm neonates with negative outcomes may be characterized with a disrupted integrity or altered permeability. Further studies in all these directions are needed, including additional investigations concerning the possible extracerebral release of S100B, and methodological problems surrounding its measurements [10,15,58].

Based on the variability of these two molecules amongst the neonatal categories, the full-term showed the lowest (in the case of S100B) or the highest (in the case of Tau) values, while no differences were observed between the preterm, IUGR and preterm-IUGR in single subgroups. Concerning the urinary NGF and cerebral volumes, and in agreement with previous findings [1,2,3], there were significant differences observed between neonates’ categories, with the preterm-IUGR newborns exhibiting the lowest values in both subgroups.

Unlike NGF, no relationship between S100B, Tau, and cerebral volumes was found, suggesting that the latter two proteins are not involved in brain growth at 30–40 days after birth. Our data also indicates that these three neurotrophic factors did not correlate with one another, however, data further suggest that the brain of subjects with negative outcomes at two years of age is characterized with an increased and a decreased/impaired activity of their glial and neuronal cells, respectively.

The diagnostic significance of the parameters considered in this study towards the early recognition of neonates at risk of impaired neurodevelopment was evaluated with the assessment of the AUCs and predictive logistic regression models. In accordance with previous data [1,3], the cerebral volumes showed excellent prognostic accuracy. Interestingly, the accuracy of the predictive logistic regression model including Tau and NGF was excellent and comparable to that shown by the cerebral volumes.

## 4. Materials and Methods

### 4.1. Neonate Population

For the present study, a group of 30–40 days old, highly selected neonates (70 in total, including 22 full-term, 17 preterm, 15 IUGR, and 16 preterm-IUGR) was recruited. Enrolment consisted of a two-phase procedure. In the first phase, the exclusion and inclusion criteria were followed. The exclusion criteria were perinatal asphyxia, intraventricular or cerebral hemorrhage, periventricular leukomalacia, sepsis, infectious diseases of the brain, invasive ventilation during the first days of life, maternal diabetes, maternal drug, tobacco and alcohol use and abuse in pregnancy, maternal pregestational and gestational obesity, and paternal obesity. The inclusion criteria were moderate prematurity [GA ≥ 32 weeks (wks)], moderate IUGR [birth weight (BW) ≥ 1500 g], full-term birth at 37–38 wks, and breastfeeding for at least 3 weeks with mother’s milk before the enrollment to rule out possible bias [59].

At this phase, urine samples were collected, and biochemical and anthropometric parameters were measured. After a two-year follow-up, the assessed subjects underwent neurodevelopment evaluations which permitted the recognition of the normal and the abnormal neurodevelopment subgroups. A further selection was performed to rule out the gender and GA biases considering that the risk of neurological damage varies with the GA or gender [60,61], and, similarly, S100B changes in the cord blood and urine at birth [24,25]. Additionally, this approach made it possible to show subtle differences between the two subgroups. Thus, 8 preterm, 8 IUGR, and 8 preterm IUGR neonates were included in the abnormal neurodevelopment subgroup (GA: 35.33 ± 1.8; sex: 7 males, 17 females), whereas 22 full-term, 9 preterm, 7 IUGR, and 8 preterm IUGR were included in the normal neurodevelopment group (GA: 36.1 ± 1.8, sex: 18 males, 28 females).

In the enrolled population, 63/70 (95.8%) of subjects were fed with mother’s milk and the remainder (3/70), who were pre-IUGR neonates, received fortified breast milk (which consisted of mother’s milk with multi-nutrient fortifier that was added to achieve the recommended nutrient intake) during their first week of life.

In the case of prematurity, the post-birth age was corrected to the equivalent age. Diagnosis of IUGR was performed using the fetal ultrasound criteria [1,2,3].

### 4.2. Urine Collection

The collection and treatment of the urine samples was performed as previously described [62].

Measurement of leukocytes and nitrite were tested with a multiple test strip (Combi-Screen PLUS, Analyticon Biotechnologies AG, Lichtenfels, Germany) to exclude possible urinary infections and local hemolysis. The biochemical parameters were expressed as a ratio to urinary creatinine to avoid differences in the urinary flow rate. Urinary creatinine (mg/dL) was measured using an enzymatic method (Advia ECREA_2, 04992596, performed on Advia 1800 analyzer Siemens, Healthcare S.r.l., Milano, Italy).

### 4.3. S100B, Tau, and NGF Assays

The urinary levels of S100B, Tau, and NGF (pg/mL) were assayed using the colorimetric ELISA kits for human S100B (Abcam, Cambridge, UK) [63], human total Tau (Thermo-Fisher Scientific, Segrate, Italy) and for human (total) beta-NGF (Thermo-Fisher Scientific) [3]. Assays were performed according to the manufacturer’s instructions. Values were expressed as a ratio to urinary creatinine (ng/mg) to avoid differences in the urinary flow rate.

### 4.4. 3D Echo-Measured Cerebral Volumes

The regional cerebral volumes were measured as previously described using the Virtual Organ Computer-Aided Analysis software (VOCAL) (Vocal II, software Item H44842LM GE ULTRASOUNDS, Chicago, IL, USA) [1,2,3]. Measurements of the cerebral volumes were obtained using a blinded sonographer showing an intra- and inter-operator variability that was <2% in the case of WBV, TV, CV, and <5% in the case of FCV.

### 4.5. Neurodevelopmental Assessment at 2 Years of Age

Neurodevelopmental assessments were performed using the Griffiths III test [1]. For this study we considered the total developmental quotient (DQ). Children were evaluated by psychologists and neonatologists with experience in neurodevelopmental examination and who had completed an accredited training course on the Griffiths scales. All of them were blinded to the data of the cerebral volumes. Using a cut-off value of 85, obtained considering the DQ mean −1SD, a DQ ≥ 85 was considered as normal. Infants with a DQ≥ or <85 were then included in the subgroups of the normal or abnormal neurodevelopment, respectively [1].

### 4.6. Statistical Analysis

Data analysis and graphs were carried out using IBM SPSS (version 23), MEDCALC, and GraphPad Prism (version 6.01) statistical software.

Biochemical tests were performed three times. Each time included at least three replicates. Results are expressed as means ± SD. Measurements of the cerebral volumes were obtained as an average of four repeated assessments. The D’Agostino–Pearson normality test was used to determine the normal distribution of variables. Comparison between two groups was performed using the non-parametric Mann–Whitney test or the unpaired *t*-test. Fisher’s exact test was performed for the sex percentages comparison. Multiple comparison was carried out using the Kruskal–Wallis test. The predictive accuracy of variables was quantified as the AUC. The ROC curve was constructed with measurements from the normal vs. abnormal neurodevelopment subgroups (with and without the full-term category of neonates). Correlations between the variables were assessed through Spearman’s rho rank correlation coefficient analysis. Graphs indicated the median and the IQR. Bivariate and multivariate analysis aiming to evaluate the ability of all predictors in determining the neurodevelopment impairment at two years of age was performed using binary logistic regression and calculations of the respective ORs with 95% CIs. Logistic regression models were then complemented with the predictive accuracy test that was quantified as the AUC. Comparison of AUCs was performed using MEDCALC software (https://www.medcalc.org/calc/, accessed on 30 May 2023).

## 5. Conclusions

The early identification of neonates and children at higher risk of neurodevelopmental impairments or disability is very important for planning clinical early interventions that are aimed at improving developmental issues [4,5].

The present study indicates that urinary S100B and Tau assessed at 30–40 days post-birth can be considered as valuable parameters for the early detection of future negative outcomes in preterm and IUGR neonates. It further supports the important role that the 3D echo cerebral volumes, particularly the TV, may independently hold in this matter [1,2,3], and preliminary identifies a predictive logistic multivariate model demonstrating an excellent predictive accuracy similar to that of the brain volumes, thereby representing a comparable and valid prognostic alternative.

Furthermore, although not elucidating the tissue specificity of urinary S100B, our results suggest that the brains of the IUGR and preterm neonates with negative outcomes may be characterized by a reduced growth and subclinical injury of their glial components or, alternatively, concurrent active responses of the glia to reduced/impaired functions of neurons.

## Figures and Tables

**Figure 1 ijms-24-11549-f001:**
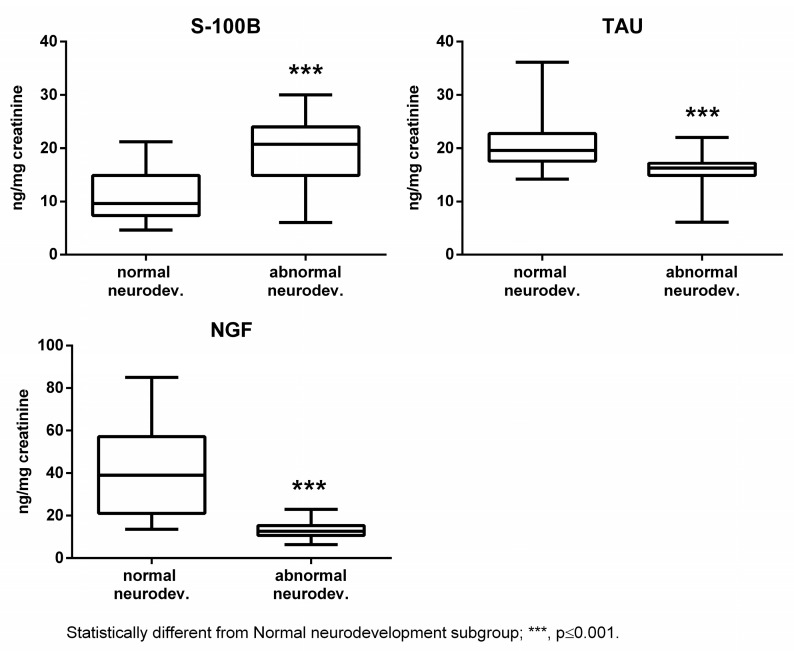
Urinary S100B, Tau, and NGF in the normal and abnormal neurodevelopment subgroups.

**Figure 2 ijms-24-11549-f002:**
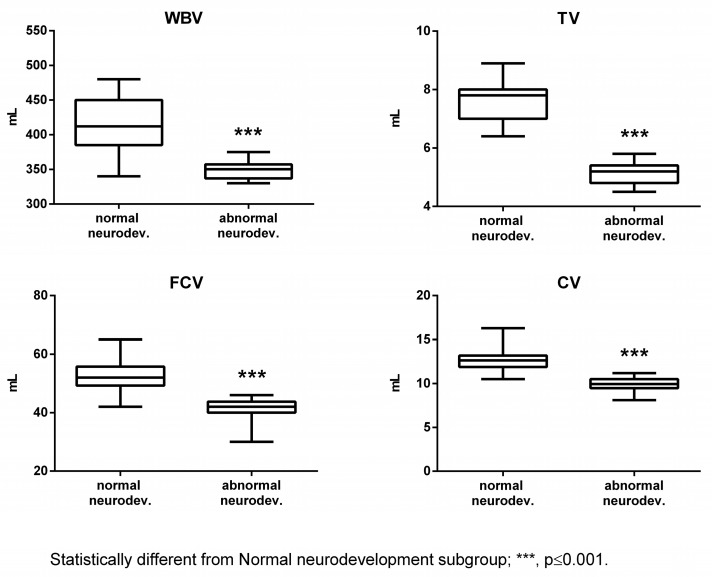
Cerebral volumes in the normal and abnormal neurodevelopment subgroups. WBV: whole brain volume; TV: thalamus volume; FCV: frontal cortex volume; and CV: cerebellum volume.

**Figure 3 ijms-24-11549-f003:**
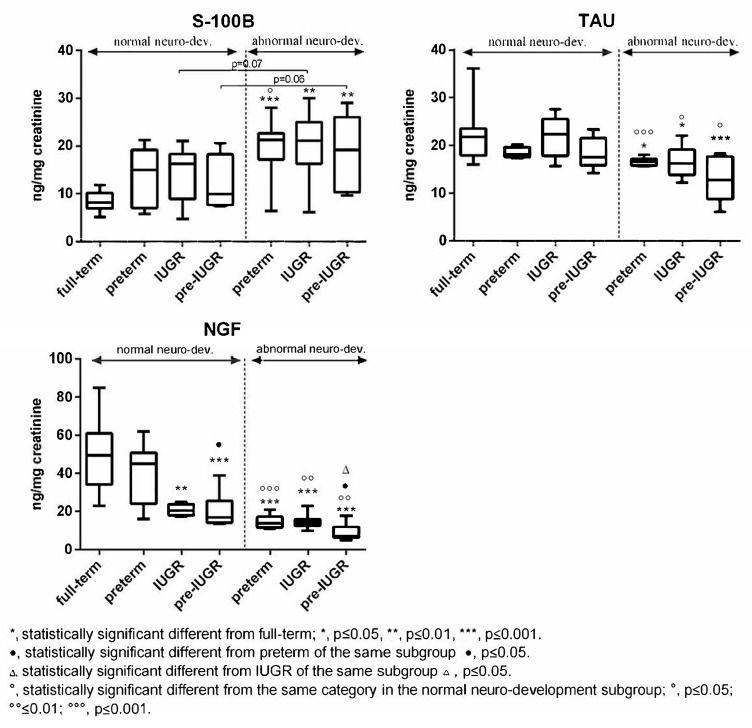
Urinary S100B, Tau, and NGF in the normal and abnormal neurodevelopment subgroups with reference to neonates’ categories.

**Figure 4 ijms-24-11549-f004:**
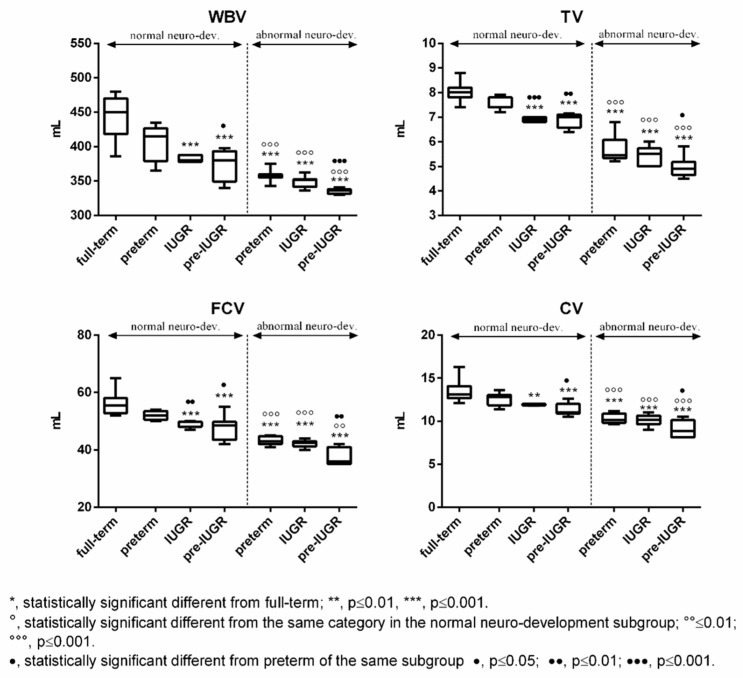
Cerebral volumes in the normal and abnormal neurodevelopment subgroups with reference to neonates’ categories. WBV: whole brain volume; TV: thalamus volume; FCV: frontal cortex volume; and CV: cerebellum volume.

**Table 1 ijms-24-11549-t001:** Variables measured at 30–40 days of the postnatal period in the whole population.

	S100B	Tau	NGF	WBV	TV	FCV	CV	BW
n	70	70	70	70	70	70	70	70
Median	1.1	1.8	2	385	7	49.5	11.9	1905
IQR	0.8–1.9	1.6–2.2	1.4–4.2	352–425	5.6–7.8	43–53.9	10–13	1650–2105
Mean ± SD	1.4 ± 0.7	1.9 ± 0.5	2.8 ± 1.9	390 ± 45.7	6.8 ± 1.3	48.7 ± 7.3	11.7 ± 1.9	1900 ± 305

WBV: whole brain volume; TV: thalamus volume; FCV: frontal cortex volume; CV: cerebellum volume; BW: birth weight; IQR: interquartile range; SD: standard deviation; and n: number of subjects examined. S100B, Tau, and NGF were expressed as ng/mg of creatinine. WBV, TV, FCV, and CV were expressed as mL, and BW was expressed as g.

**Table 2 ijms-24-11549-t002:** Variables measured at 30–40 days of the postnatal period in full-term, preterm, IUGR, and preterm-IUGR subjects.

	Tau	S100B	NGF	WBV	TV	FCV	CV
	**Full-term**						
n	22	22	22	22	22	22	22
Median	2.2	0.82	5	450	8	56	13.1
IQR	1.7–2.3	0.69–1.01	3.2–6.2	418–470	7.8–8.5	53–59	12.7–14.1
Mean ± SD	2.2 ± 0.5	0.83 ± 0.19	5 ± 1.8	440 ± 32	8.13 ± 0.48	56 ± 3.5	13.5 ± 1
	**Preterm**						
n	17	17	17	17	17	17	17
Median	1.7 *	1.88 ***	1.9 **	366 **	7.2 ***	50 ***	11.4 **
IQR	1.6–1.8	1.2–2.17	1.3–4.85	358–420	5.3–7.8	42–52	10.1–12.85
Mean ± SD	1.7 ± 0.13	1.7 ± 0.65	2.7 ± 1.8	383.5 ± 31	6.53 ± 1.2	47.5 ± 5	11.5 ± 1.3
	**IUGR**						
n	15	15	15	15	15	15	15
Median	1.78	1.83 ***	1.5 ***	362 ***	5.5 ***	46 ***	10.7 ***
IQR	1.6–2.23	1.49–2.15	1.3–2	352–380	5–6.9	42.2–48	10–11.9
Mean ± SD	1.9 ± 0.5	1.77 ± 0.7	1.6 ± 0.43	364 ± 18	6 ± 0.8	45.5 ± 3.5	10.8 ± 1
	**Preterm-IUGR**						
n	16	16	16	16	16	16	16
Median	1.6 **	1.3 *	1.36 ***	340 ***	6.1 ***	44 ***	10.5 ***
IQR	1.1–1.9	0.94–2.03	0.71–1.84	335.5–382.5	4.8–7	35.2–48.7	8.3–11
Mean ± SD	1.6 ± 0.4	1.54 ± 0.7	1.4 ± 0.82	354.3 ± 25	5.6 ± 1.1	42.6 ± 7.5	10.2 ± 1.5

WBV: whole brain volume; TV: thalamus volume; FCV: frontal cortex volume; CV: cerebellum volume; IQR: interquartile range; SD: standard deviation; and n: number of subjects examined. Tau, S100B, and NGF were expressed as ng/mg of creatinine. WBV, TV, FCV, and CV were expressed as mL. Statistically significant different from full-term; *, *p* ≤ 0.05; **, *p* ≤ 0.01, ***, *p* ≤ 0.001.

**Table 3 ijms-24-11549-t003:** ROC curves of the individual parameters and the results of bivariate and multivariate logistic regression analysis predicting the probability of impaired neurodevelopment at two years of age.

Predictor	ROC Curve AUC (95% CI)	Bivariate Logistic Regression OR (95% CI); *p*-Value
S100B	0.845	1.24 (1.15–1.45); *p* < 0.001
Tau	0.833	0.649 (0.497–0.847); *p* = 0.001
NGF	0.958	0.664 (0.529–0.834); *p* < 0.001
WBV *	0.969	0.875 (0.815–0.938); *p* < 0.001
TV *	1.00	<0.001; *p* < 0.001
FCV *	0.980	0.447 (0.294–0.680); *p* < 0.001
CV *	0.984	0.02 (0.01–0.27); *p* = 0.003
**Multivariate Logistic Regression Models**		
**Predictors**	**ROC Curve** **AUC (95% CI)**	
S100B + Tau	0.959 (0.899–1.0); *p* < 0.001	
NGF + Tau	0.994 (0.983–1.0); *p* < 0.001	

WBV: whole brain volume; TV: thalamus volume; FCV: frontal cortex volume; CV: cerebellum volume; AUC = area under the curve; CI = confidence interval; and OR = odds ratio. * Not included in the multivariate model because of its excellent accuracy.

## Data Availability

All data, materials and our knowledge related to this work are available from the corresponding author MCA upon request.

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
