# Peer review of "Biochemical and Anthropometric Parameters for the Early Recognition of the Intrauterine Growth Restriction and Preterm Neonates at Risk of Impaired Neurodevelopment"

_ijms, 2023, doi:10.3390/ijms241411549_

Round 1

Reviewer 1 Report

The paper entitled “Further insights into the early recognition of IUGR and pre-term neonates at risk of impaired neurodevelopment later in life” is a well-written study reporting that urinary S100B and Tau assessed at 30-40 days of the postnatal period could be new and useful parameters for the early detection of future neurodevelopmental abnormalities in preterm and Intrauterine growth restriction (IUGR) neonates. The study includes a group of 30-40 days old neonates (70 in total, including 22 full-term, 17 preterm, 15 IUGR, and 16 preterm-IUGR). Inclusion and exclusion criteria are reported as well as the number/ID of ethics approval. The work is well-written and structured. The methods are clearly described, and the results are of clear importance in the field. 

Author Response

Thank you to Reviewer 1 for the comments.

Reviewer 2 Report

This interesting work focuses on the possible association between the two proteins, S100B and Tau, with neurodevelopmental delay in life in preterm subjects and with intrauterine growth restriction (IUGR). The authors correlated urinary S100B and Tau levels, measured at 30-40 days of the postnatal period, with motor and behavioral data over a two-year follow-up.

The manuscript could be improved by providing additional details, addressing a mechanism or introducing an hypothesis, and by adding a limitation section. 

in particular, the discussion could be expanded by including some commnets on some kind of hypothetical explanations regarding the origin of the protein(s) or considering the putative mechanisms underlying their release in the blood, cerebrospinal fluid, as well as the final detection in the urine. (Creatinine ratio is fine, but it is just a minimun requirement).

A paragraph can be added with the limitations of the study, such as technical aspects and possible artifacts, the presence of possible confounding factors, the specificity and predictive significance of the observed association additional issues or parameters that would be lacking, etc.

Furthermore, it should be considered that S100B is detected and/or could be released from various tissues, it is present in serum and has been related to inflammatory processes, finally, what is the specificity of the test compared to other proteins of the S100 family?

S100B as other factors are present in milk/breast feading; may this issue represent an interfering factor or a confounding factor related to the assumption and or alterations in the gut barrier? 

Did the Author consider any evaluation of the breast feading parameters in this study population of newborns? And/or were measured the levels of S100B in maternal milk? What about other factors e.g. BDNF, and GDNF? (Consider they were detected in breast milk, too).

Multivariate statistical analysis involving other blood or serum parameters may contribute to support and strenghten the results as well as provide hints for proposing possible mechanisms or hypothesis or tools and perspectives for further studies. 

In line 151, table 2 is missing the caption.

The conclusion section may have to be moved before the materials and methods as indicated in the instruction for the journal, please verify.

fine

Author Response

We thank Reviewer 2 for his/her right comments.

In agreement with his/her suggestions we have now amended the text and used the template of JCM.

Firstly, we performed a bivariate and multivariate logistic regression and AUCs comparison.

We added a new table with the results found.

Although preliminary, data were very indicative in showing the excellent prognostic accuracy of the parameters studied. Consequently, we updated the title to “Biochemical and anthropometric parameters for the early recognition of IUGR and preterm neonates at risk of impaired neurodevelopment”, to emphasize that.

In the various sections of the manuscript, we have elucidated some important details concerning S100B as follows.

  • We have now specified that we recruited neonates who were breastfed with mother’s milk for at least 3 weeks  before the enrolment to limit possible bias (such population is also under study for renal investigation for which the use of formulae-milk is excluded) and that only 3/70 subjects were fed with fortified mother’s milk during the first week of life.
  • We formulated some hypotheses about possible mechanisms underlying the data obtained.
  • We also mentioned the limits concerning the real source of urinary S100B and methodological problems surrounding the measurement of S100B.